# Using Action-Level Metrics to Report the Performance of Multi-Step Keyboards

Gulnar Rakhmetulla*
Human-Computer Interaction Group
University of California, Merced

Ahmed Sabbir Arif†
Human-Computer Interaction Group
University of California, Merced

Steven J. Castellucci‡
Independent Researcher
Toronto

I. Scott MacKenzie§
Electrical Engineering and Computer Science
York University

Caitlyn E. Seim¶
Mechanical Engineering
Stanford University

## ABSTRACT

Computer users commonly use multi-step text entry methods on handheld devices and as alternative input methods for accessibility. These methods are also commonly used to enter text in languages with large alphabets or complex writing systems. These methods require performing multiple actions simultaneously (chorded) or in a specific sequence (constructive) to produce input. However, evaluating these methods is difficult since traditional performance metrics were designed explicitly for non-ambiguous, uni-step methods (e.g., QWERTY). They fail to capture the actual performance of a multi-step method and do not provide enough detail to aid in design improvements. We present three new action-level performance metrics: UnitER, UA, and UnitCX. They account for the error rate, accuracy, and complexity of multi-step chorded and constructive methods. They describe the multiple inputs that comprise typing a single character—action-level metrics observe actions performed to type one character, while conventional metrics look into the whole character. In a longitudinal study, we used these metrics to identify probable causes of slower text entry and input errors with two existing multi-step methods. Consequently, we suggest design changes to improve learnability and input execution.

**Index Terms:** Human-centered computing—Empirical studies in HCI; Human-centered computing—Usability testing; Human-centered computing—User studies

## 1 INTRODUCTION

Most existing text entry performance metrics were designed to characterize uni-step methods that map one action (a key press or a tap) to one input. However, there are also multi-step constructive and chorded methods. These require the user to perform a predetermined set of actions either in a specific sequence (pressing multiple keys in a particular order) or simultaneously (pressing multiple keys at the same time). Examples of constructive methods include multi-tap [16] and Morse code [19,38]. Chorded methods include braille [3] and stenotype [21]. Many text entry methods used for accessibility and to enter text in languages with large alphabets or complex writing systems are either chorded or constructive. However, current metrics for analyzing the performance of text entry techniques were designed for uni-step methods, such as the standard desktop keyboard. Due to the fundamental difference in their input process, these metrics often fail to accurately illustrate participants'

---

*e-mail: grakhmetulla@ucmerced.edu

†e-mail: asarif@ucmerced.edu

‡e-mail: stevenc@cse.yorku.ca

§e-mail: mack@cse.yorku.ca

¶e-mail: cseim@stanford.edu

Table 1: Conventional error rate (ER) and action-level unit error rate (UnitER) for a constructive method (Morse code). This table illustrates the phenomenon of using Morse code to enter *"quickly"* with one character error (*"l"*) in each attempt. ER is 14.28% for each attempt. One of our proposed action-level metrics, UnitER, gives a deeper insight by accounting for the entered input sequence. It yields an error rate of 7.14% for the first attempt (with two erroneous dashes), and an improved error rate of 3.57% for the second attempt (with only one erroneous dash). The action-level metric shows that learning has occurred with a minor improvement in error rate, but this phenomenon is omitted in the ER metric, which is the same for both attempts.

| | | ER (%) | UnitER (%) |
|---|---|---|---|
| Presented Text | q u i c k **l** y | | |
| Transcribed Text | q u i c k **j** y | 14.28 | 7.14 |
| Presented Text | q u i c k **l** y | | |
| Transcribed Text | q u i c k **p** y | 14.28 | 3.57 |

actions in user studies evaluating the performance of multi-step methods.

To address this, we present a set of revised and novel performance metrics. They account for the multi-step nature of chorded and constructive text entry techniques by analyzing the actions required to enter a character, rather than the final output. We posit that while conventional metrics are effective in reporting the overall speed and accuracy, a set of action-level metrics can provide extra details about the user's input actions. For designers of multi-step methods, this additional insight is crucial for evaluating the method, identifying its pain points, and facilitating improvements. More specifically, conventional error rates fail to capture learning within chords and sequences. For instance, if entering a letter requires four actions in a specific order, with practice, users may learn some of these actions and the corresponding sub-sequences. Conventional metrics ignore partially correct content by counting each incorrect letter as one error, giving the impression that no learning has occurred. UnitER, in contrast, accounts for this and provides designers with an understanding of (1) whether users learned some actions or not and (2) which actions were difficult to learn, thus can be replaced (Table 1).

The remainder of this paper is organized as follows. We start with a discussion on the existing, commonly used text entry performance metrics, and then elaborate on our motivations. This is followed with a set of revised and new performance metrics targeted at multi-step input methods. We proceed to validate the metrics in a longitudinal study evaluating one constructive and one chorded input methods. The results demonstrate how the new metrics provide a

deeper insight into action-level interactions. Researchers identify factors compromising the performance and learning of a multi-step keyboard, and suggest changes to address the issues.

## 2 RELATED WORK

Conventional metrics for text entry speed include characters per second (CPS) and words per minute (WPM). These metrics represent the number of resulting characters entered, divided by the time spent performing input. Text entry researchers usually transform CPS to WPM by multiplying by a fixed constant (60 seconds, divided by a word length of 5 characters for English text entry) rather than recalculating the metrics [8].

A common error metric, keystrokes per character (KSPC), is the ratio of user actions, such as keystrokes, to the number of characters in the final output [24, 39]. This metric was designed primarily to measure the number of attempts at typing each character accurately [24]. However, many researchers have used it to represent a method's potential entry speed as well [25, 42], since techniques that require fewer actions are usually faster. Some researchers have also customized this metric to fit the need of their user studies. The two most common variants of this metric are gesture per character (GPS) [10, 45, 46] and actions per character (APC) [45], which extend keystrokes to include gestures and other actions, respectively. Error rate (ER) and minimum string distance (MSD) are metrics that measure errors based on the number of incorrect characters in the final output [8]. Another metric, erroneous keystrokes (EKS) [8], considers the number of incorrect actions performed in an input episode. None of these metrics considers error correction efforts, thus Soukoreff and MacKenzie [39] proposed the total error rate (TER) metric that combines two constituent errors metrics: corrected error rate (CER) and not-corrected error rate (NER). The former measures the number of corrected errors in an input episode and the latter measures the number of errors left in the output.

Accessibility text entry systems mainly use constructive or chorded techniques. Both techniques require the user to perform multiple actions for one input, but the difference is in the order of actions. Constructive techniques require the user to perform a combination of actions *sequentially* to enter one character. The chorded techniques require the user to press multiple keys *simultaneously* to enter one character, like playing a chord on a piano. Morse code is a constructive text entry system that was named one of the "Top 10 Accessibility Technologies" by RESNA [33]. In 2018, Android phones integrated this entry system as an accessibility feature for users with motor impairments [43]. Morse code users can tap on one or two keys to enter short sequences of keystrokes representing each letter of the alphabet. This method reduces the dexterity required to enter text, compared to the ubiquitous QWERTY keyboard. Braille is a tactile representation of language used by individuals with visual impairments. Braille keyboards [28, 29, 41] contain just six keys so that users do not need to search for buttons; instead, users can keep their hands resting across the keys. To produce each letter using this system, users must press several of these keys simultaneously.

| Word | কান্ড | |
|---|---|---|
| Conjunct | কা | ন্ড |
| Primary Character | ক + আ | ন + ড |

Figure 1: Text entry methods for languages that have a large alphabet or a complex writing system usually use constructive or chorded techniques. This figure breaks down a Bangla word to its primary characters.

Text entry methods for languages that have a large alphabet or a complex writing system also use constructive or chorded techniques. The Bangla language, for example, has approximately 11 vowels,

Table 2: Performance metrics used to evaluate chorded and constructive keyboards in recent user studies. "ALM" represents action-level metric.

| | Technique | Speed | Accuracy | ALM | $R^2$ |
|---|---|---|---|---|---|
| Chorded | Twiddler [23] | WPM | ER | *NA* | Yes |
| | BrailleType [29] | WPM | MSD ER | *NA* | *NA* |
| | ChordTap [44] | WPM | ER | *NA* | Yes |
| | Chording Glove [34] | WPM | MSD ER | *NA* | *NA* |
| | Two-handed [47] | WPM | ER | *NA* | *NA* |
| Construct. | Mutitap [15] | WPM | ER | *NA* | *NA* |
| | Reduced QWERTY [17] | WPM | *NA* | *NA* | *NA* |
| | JustType [27] | WPM | *NA* | *NA* | *NA* |
| | UOIT [2] | WPM | TER | *NA* | *NA* |

36 consonants, 10 inflexions, 4 signs, and 10 numeric characters. Additionally, Bangla supports combinations between consonants and consonant and diacritic forms of the vowels. In Fig. 1, the top row presents one Bangla word, pronounced kăn-dō, meaning "stem", composed of two conjoined letters. The first is a combination of a consonant and a vowel, while the second is a combination of two consonants. Due to the large alphabet, Bangla writing systems map many characters to each key of the QWERTY layout, thus requiring the user to disambiguate the input using either the constructive or the chorded techniques. Combining two or more characters also requires performing a combination of actions sequentially or simultaneously. Sarcar et al. [35] proposed a convention for calculating the most common performance metrics for such techniques that considers both the input and the output streams.

Little work has focused on performance metrics for such multi-step input techniques. Grossman et al. [18] defined "soft" and "hard" errors for two-level input techniques, where errors at the first level are considered "soft" and errors at the second level are considered "hard".[1] Seim et al. [36] used a dynamic time warping (DTW) algorithm [20] to measure the accuracy of chorded keystrokes on a piano keyboard. Similar to the Levenshtein distance [22], it measures the similarity between two sequences, but accounts for variants in time or speed. Arif and Stuerzlinger [9] proposed a model for predicting the cost of error correction with uni-step techniques by breaking down each action into low-level motor and cognitive responses. In a follow-up work, Arif [4] discussed how the model can be extended to multi-step techniques and to construct new metrics, however did not develop the metrics or validated them in user studies.

Based on Shannon's information theory [37] and the observation that the speed-accuracy trade-off arises as a predictable feature of communication within humans [40], a method-independent throughput metric was proposed, where the amount of information transmitted via a text entry method per unit time reflects the input efficiency of the method [49].

Some have also used custom conventions for keyboard optimization. Bi et al. [11] optimized a keyboard layout to reduce stylus travel distance for multiple languages. Oulasvirta et al. [30] minimized thumb travel distance for two-thumb text entry on mobile devices. Rakhmetulla and Arif [32] optimized a smartwatch keyboard to facilitate the transference of gesture typing skill between devices by maintaining similarities between the gestures drawn on them. Some have also proposed new metrics to capture the effectiveness of predictive features, such as auto-correction and word prediction [6], and developed tools to enable action-level analysis of input[2]. Table 2

---

[1] With two-level techniques, users usually perform two actions to select the desired character, for example, the first action to specify the region and the second to choose the character.

[2] A web application to record text entry metrics, `https://www.asarif.com/resources/WebTEM`

Table 3: Performance metrics used to evaluate *"x"* for chorded (Twiddler) and constructive (Morse) methods.

| Technique | Correct input for *'x'* | Actual input for *'x'* | ER | UnitER |
|---|---|---|---|---|
| Twiddler [23] | M**R0**0 | M**0R**0 | 100 | 50 |
| Morse code [38] | -..**-** | -..**.** | 100 | 25 |

shows the performance metrics used to evaluate recent chorded and constructive text entry techniques.

## 3 MOTIVATION

### 3.1 Partial Errors and Predictions

Users can make partial errors in the process of performing a chord or a sequence of actions. To enter *"x"*, a Twiddler [23] user could incorrectly perform the chord *"M0R0"* instead of *"MR00"*, and a Morse code user could incorrectly perform the sequence *"-..."* instead of *"-..-"* (Table 3). In both cases, user actions would be partially correct. However, typical error metrics ignore this detail by considering the complete sequence as one incorrect input. Hence, they yield the same value as when users enter no correct information. In reality, users may have learned, and made fewer mistakes within a chord or a sequence. Not only does this show an incomplete picture of user performance, it also fails to provide the means to fully explore learning of the text entry method. More detailed metrics of multi-step keyboard interaction can facilitate improved input method design through a better understanding of the user experience. These data can also train algorithms to predict and compensate for the most common types of errors.

### 3.2 Correction Effort

Prior research [5, 9] shows that correction effort impacts both performance and user experience, but most current error metrics do not represent the effort required to fix errors with constructive techniques. With uni-step character-based techniques, one error traditionally requires two corrective actions: a backspace to delete the erroneous character, and a keystroke to enter the correct one [9]. Suppose two users want to input *"-..-"* for the letter *"x"*. One user enters *"-..."*, the other enters *"...-"*. Existing error metrics consider both as one erroneous input. However, correcting these may require different numbers of actions. If a technique allowed users to change the direction of a gesture mid-way [7], the errors would require one and five corrective actions, respectively. If the system only allowed the correction of one action at a time, then the former would require two and the latter would require five corrective actions. In contrast, if the system does not allow the correction of individual actions within a sequence, both would require five corrective actions. Hence, error correction relies on the correction mechanism of a technique, the length of a sequence, and the position and type of error (insertion, deletion, or substitution). Existing metrics fail to capture this vital detail in both chorded and constructive text entry techniques.

### 3.3 Deeper Insights into Limitations

The proposed metrics aim to give quantitative insights into the learning and input of multi-step text entry techniques. Insights, such as tapping locations affecting speed, might seem like common sense, but action-level metrics can also provide similar insights for less straight-forward interactions, such as breath puffs, tilts, swipes, etc. Although some findings might seem unsurprising for experts, the proposed metrics will benefit the designers of new techniques aimed at multi-step text entry and complement their efforts and insights.

The new metrics can facilitate design improvements by quantifying the design choices. Conventional metrics identify difficult and erroneous letters, while our UnitER, UA, and UnitCX indicate what is contributing towards it. The action-level metrics can help us

to identify difficult-to-enter sequences, so they can be assigned to infrequent characters or avoided altogether. For example, UnitER and UA can reveal target users having difficulty performing the long-press of a three-action input sequence for some character. Designers then can replace the long-press with an easier action (e.g., a physical button press) for a particular group of users to reduce that letter's error rate.

## 4 NOTATIONS

In the next section, we propose three new action-level metrics for evaluating multi-step text entry techniques. For this, we use the following notations.

- Presented text ($PT$) is the text presented to the user for input, $|PT|$ is the length of $PT$, $PT_i$ is the $i$th character in PT, $pt_i$ is the sequence of actions required to enter the $i$th character in $PT$, and $|pt_i|$ is the length of $pt_i$.

- Transcribed text ($TT$) is the text transcribed by the user, $|TT|$ is the length of $TT$, $TT_i$ is the $i$th character in $TT$, $tt_i$ is the sequence of actions performed by the user to enter the $i$th character in $TT$, and $|tt_i|$ is the length of $tt_i$.

- Minimum string distance (MSD) measures the similarity between two sequences using the Levenshtein distance [22]. The "distance" is defined as the minimum number of primitive operations (*insertion, deletion, and substitution*) needed to transform a given sequence ($TT$) to the target sequence ($PT$) [24].

- Input time ($t$) is the time, in seconds, the user took to enter a phrase ($TT$).

- An action is a user action, including a keystroke, gesture, tap, finger position or posture, etc. Action sequence (AS) is the sequence of all actions required to enter the presented text. $|AS|$ is the length of AS. We consider all sub-actions within a chord as individual actions. For example, if a chord requires pressing three keys simultaneously, then it is composed of three actions.

- We denote a substring (suffix) of a string $s$ starting at the $k$th character as $s[k :]$. For example, if $s =$ "quickly", substring $s[1 :]$ is string "uickly".

## 5 SPEED AND ACCURACY METRICS

### 5.1 Inputs per Second (IPS)

We present IPS, a variant (for convenience of notation) of the commonly used CPS metric [46] to measure the entry speeds of multi-step techniques.

$$\text{IPS} = \frac{|AS|}{t} \qquad (1)$$

IPS uses the length of the action sequence $|AS|$ instead of the length of transcribed text $|TT|$ to account for all multi-step input actions. It is particularly useful to find out if the total number of actions needed for input is affecting performance or not.

### 5.2 Actions per Character (APC)

For convenience of notation, we present Actions per Character (APC), a variant of the Keystrokes per Character (KSPC) [8, 24], and the Gesture per Character (GPS) [10, 39] metrics. It measures the average number of actions required to enter one input unit, such as a character or a symbol [46]. One can measure accuracy by comparing the number of actions required to enter presented text to the number of actions actually performed by participants.

$$\text{APC} = \frac{|AS|}{|TT|} \qquad (2)$$

## 6 PROPOSED ACTION-LEVEL METRICS

### 6.1 Unit Error Rate (UnitER)

The first of our novel metrics, unit error rate (UnitER) represents the average number of errors committed by the user when entering one input unit, such as a character or a symbol. This metric is calculated in the following three steps:

#### 6.1.1 *Step 1: Optimal Alignment*

First, obtain an optimal alignment between the presented ($PT$) and transcribed text ($TT$) using a variant of the MSD algorithm [26]. This addresses all instances where lengths of presented ($|PT|$) and transcribed text ($|TT|$) were different.

$$\text{MSD}(a,b) = \begin{cases} |b|, & \text{if } a = \text{""} \\ |a|, & \text{if } b = \text{""} \\ 0, & \text{if } a = b, \\ S \end{cases} \quad (3)$$

where $S$ is defined as

$$S = \min \begin{cases} \text{MSD}(a[1:],b[1:]) & \text{if } a[0] = b[0], \\ \text{MSD}(a[1:],b)+1, \\ \text{MSD}(a,b[1:])+1, \\ \text{MSD}(a[1:],b[1:])+1. \end{cases} \quad (4)$$

If multiple alignments are possible for a given MSD between two strings, select the one with the least number of insertions and deletions. If there are multiple alignments with the same number of insertions and deletions, then select the first such alignment. For example, if the user enters *"qucehkly"* ($TT$) instead of the target word *"quickly"* ($PT$), then $\text{MSD}(PT,TT) = 3$ and the following alignments are possible:

```
quic--kly    quic-kly    qui-ckly    qu-ickly
qu-cehkly    qucehkly    qucehkly    qucehkly
```

Here, a dash in the top sequence represents an *insertion*, a dash in the bottom represents a *deletion*, and different letters in the top and bottom represents a *substitution*. Our algorithm selects the highlighted alignment.

#### 6.1.2 *Step 2: Constructive vs. Chorded*

Because the sequence of performed actions is inconsequential in chorded methods, sort both the required ($pt_i$) and the performed actions ($tt_i$) using any sorting algorithm to obtain consistent MSD scores. Action sequences are not sorted for constructive methods since the order in which they are performed is vital for such methods.

#### 6.1.3 *Step 3: Minimum String Distance*

Finally, apply the MSD algorithm [26] to measure the minimum number of actions needed to correct an incorrect sequence.

$$\text{UnitER} = \frac{\sum_{i=1}^{|\overline{TT}|} \frac{\text{MSD}(pt_i,tt_i)}{\max(|pt_i|,|tt_i|)}}{|\overline{TT}|} \times 100\% \quad (5)$$

Here, $|\overline{TT}|$ is the length of the aligned transcribe text (same as $|\overline{PT}|$), and $\overline{pt_i}$ and $\overline{tt_i}$ are the sequence of actions (sorted for chorded techniques in *Step 2*) required and performed, respectively, to enter the $i$th character in $TT$.

This step requires certain considerations about the presence of *insertion* and *deletion* in the optimal alignment. If the $i$−th character of the aligned presented text $|\overline{TT}|$ has a deletion, then MSD of the corresponding $i$−th character is 100%. But when $|\overline{PT}|$ has an *insertion*, a misstroke error is assumed, as it is the most common cause of *insertions* [14]. A misstroke errors occur when the user mistakenly strokes (or taps) an incorrect key. However, the question remains: To

which character do we attribute the insertion? For this, we propose comparing the MSD-s of current $TT_i$ to the neighboring letters of $PT_i$ (which are different for different layouts), and attributing it to the one with the lowest MSD. If there is a tie, attributed it to the right neighbor $PT_i + 1$. We propose this simplification, since it is difficult to determine the exact cause of an *insertion* in such a scenario.

This metric can be used to measure error rate of a specific character, in which case, Equation 6 considers only the character under investigation, where $c$ is the character under investigation and $Total(c)$ is the total occurrence of $c$ in the transcribed text.

$$\text{UnitER}(c) = \frac{\sum_{i=1}^{|\overline{TT}|} \frac{\text{MSD}(pt_i,tt_i)}{\max(|pt_i|,|tt_i|)} \text{(if } PT_i = c)}{Total(c)} \quad (6)$$

### 6.2 Unit Accuracy (UA)

Unit accuracy (UA) is the opposite and simply a convenient reframing of UnitER. Instead of error rate, UA represents the accuracy rate of a unit. Also, unlike UnitER, the values of UA range between 0 and 1 inclusive to reflect the action-level nature of the metric (i.e., 0%–100%). UA can be used for a specific character $c$ as well, using Equation 8.

$$\text{UA} = \frac{100 - \text{UnitER}}{100} \quad (7)$$

$$\text{UA}(c) = \frac{100 - \text{UnitER}(c)}{100} \quad (8)$$

### 6.3 Unit Complexity (UnitCX)

Apart from speed and accuracy, we also propose the following novel metric to measure an input unit's complexity. For this, each action in a unit (such as a character or a symbol) is categorized into different difficulty levels, represented by the continuous values from 0 to 1.

$$\text{UnitCX} = \frac{\left( \sum_{n=1}^{|tt_i|} \frac{d(a_n)}{|TT|} \right) - d_{\min}}{d_{\max} - d_{\min}} \quad (9)$$

Here, $d(a_n)$ signifies the difficulty level of the $n$−th action in $tt_i$, and $d_{\min}$ and $d_{\max}$ are the minimum and maximum difficulty levels of all actions within or between text entry techniques. This yields a normalized unit complexity value, ranging from 0 to 1. The difficulty level of an action is based on a custom convention, considering the posture and ergonomics, memorability, and the frequency of the letters [13]. However, more sophisticated methods are available in the literature [11, 48].

## 7 EXPERIMENT: VALIDATION

We validated the effectiveness of our proposed metrics by applying them to data collected from a longitudinal user study. This study evaluated one constructive and one chorded text entry technique. Although we conducted a comparative study, our intent was to demonstrate how the proposed metrics can provide deeper quantitative insights into the multi-step techniques' performance and limitations, specifically with respect to learning, rather than comparing the performance of the two techniques.

## 8 APPARATUS

We used a Motorola Moto G$^5$ Plus smartphone (150.2 × 74 × 7.7 mm, 155 g) at 1080 × 1920 pixels in the study (Fig. 4). The virtual multi-step keyboards used in the study were developed using the default Android Studio 3.1, SDK 27. The keyboards logged all user actions with timestamps and calculated all performance metrics directly.

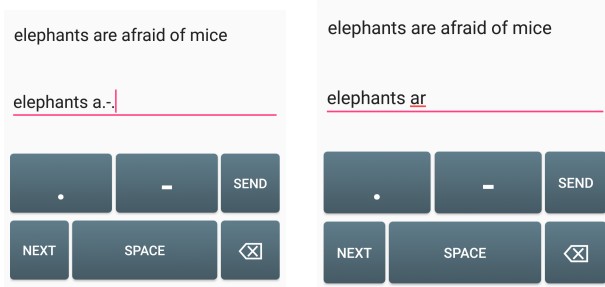

Figure 2: The Morse code inspired constructive keyboard used in the experiment.

## 9 CONSTRUCTIVE METHOD: MORSE CODE

We received the original code from the authors of a Morse code keyboard [19] to investigate the performance of a constructive keyboard. It enables users to enter characters using sequences of dots (.) and dashes (-) [38]. The keyboard has dedicated keys for dot, dash, backspace, and space (Fig. 2). To enter the letter "R", represented by ".-." in Morse code, the user presses the respective keys in that exact sequence, followed by the SEND key, which terminates the input sequence. The user presses the NEXT key to terminate the current phrase and progress to the next one.

## 10 CHORDED METHOD: SENORITA

We received the original code from the authors of the chorded keyboard Senorita [31]. It enables users to enter characters using eight keys (Fig. 3). The most frequent eight letters in English appear on the top of the key labels, and are entered with only one tap of their respective key. All other letters require simultaneous taps on two keys (with two thumbs). For example, the user taps on the "E" and "T" keys together to enter the letter "H". The keyboard provides visual feedback to facilitate learning. Pressing a key with one thumb highlights all available chords corresponding to that key, and right-thumb keys have a lighter shade than left-thumb keys (Fig. 3b).

## 11 PARTICIPANTS

Ten participants, aged from 18 to 37 years ($M = 26.1$, $SD = 5.6$), took part in the experiment. Six identified as male, four as female. None identified as non-binary. Eight were right-handed, one was left-handed, and the other was ambidextrous. They all used both hands to hold the device and their thumbs to type. All participants were proficient in English. Six rated themselves as *Level 5: Functionally Native* and four as *Level 4: Advanced Professional* on the Interagency Language Roundtable (ILR) scale [1]. All participants were experienced smartphone users, with an average of 9.3 years' experience ($SD = 1.8$). None of them had prior experience with any chorded methods, but eight had used a constructive method in the past (either multi-tap or pinyin). But none of the participants had experience with the constructive or chorded methods used in the study. They all received US $50 for volunteering.

## 12 DESIGN

We used a within-subjects design, where the independent variables were input method and session. The dependent variables were the IPS, APC, ER, and UnitER metrics. In summary, the design was as follows.

10 participants ×
5 sessions (different days) ×
2 input methods (constructive vs. chorded, counterbalanced) ×

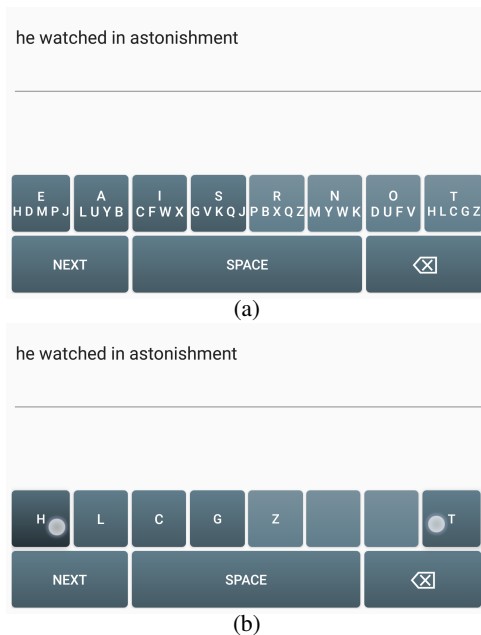

(a)

(b)

Figure 3: The Senorita chorded keyboard used in the experiment. It enables users to enter text by pressing two keys simultaneously using the thumbs. (b) Pressing a key highlights all available chords for the key.

10 English pangrams
= 1,000 phrases, in total.

## 13 PROCEDURE

To study learning all letters of the English alphabet, we used the following five pangrams during the experiment, all in lowercase.

quick brown fox jumps over the lazy dog
the five boxing wizards jump quickly
fix problem quickly with galvanized jets
pack my red box with five dozen quality jugs
two driven jocks help fax my big quiz

Participants were divided into two groups, one started with the constructive method and the other started with the chorded method. This order was switched on each subsequent session. Sessions were

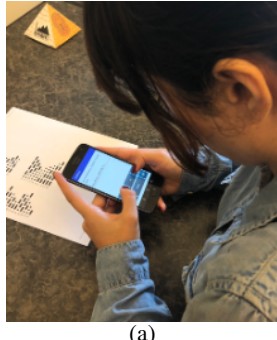
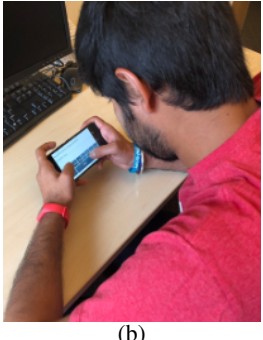

(a)                    (b)

Figure 4: Two volunteers entering text using: (a) the Morse code constructive keyboard with the assistance of a cheat-sheet, and (b) the Senorita chorded keyboard.

scheduled on different days, with at most a two-day gap in between. In each session, participants entered one pangram ten times with one method, and then a different pangram ten times with the other method. Pangrams were never repeated for a method. There was a mandatory 30–60 minutes break between the conditions to mitigate the effect of fatigue.

During the first session, we demonstrated both methods to participants and collected their consent forms. We asked them to complete a demographics and mobile usage questionnaire. We allowed them to practice with the methods, where they entered all letters (A–Z) until they felt comfortable with the methods. Participants were provided with a cheat-sheet for Morse code that included all combinations (Fig. 4a) as Morse code relies on memorization of those. For Senorita, we did not provide a cheat-sheet as the keyboard provides visual feedback by displaying the characters (i.e., it has a "built-in cheat sheet"). The experiment started shortly after that, where a custom app presented one pangram at a time, and asked participants to enter it "as fast and accurately as possible". Once done, they pressed the NEXT key to re-enter the phrase. Logging started from the first tap and ended with the last. Error correction was intentionally disabled to exclude correction efforts from the data to observe the UnitER metric. Subsequent sessions used the same procedure, excluding practice and questionnaire.

## 14 RESULTS

A Shapiro-Wilk test and a Mauchly's test revealed that the assumption of normality and sphericity were not violated for the data, respectively. Hence, we used a repeated-measures ANOVA for all analysis.

### 14.1 Inputs per Second (IPS)

The average IPS for the constructive method was 1.29 ($SD = 0.38$), and for the chorded it was 1.02 ($SD = 0.29$). For the constructive method, IPS increased from 1.02 in session one to 1.49 in session five. This effect was statistically significant ($F_{4,36} = 17.16, p < .0001$). For the chorded method, IPS increased from 0.86 in session one to 1.18 in session five. This effect was also statistically significant ($F_{4,36} = 28.26, p < .0001$). Fig. 5a displays IPS for both methods in all sessions. WPM metric also increased throughout all five sessions, which was statistically significant for both the constructive method ($F_{4,36} = 17.32, p < .0001$) and the chorded method ($F_{4,36} = 27.69, p < .0001$).

### 14.2 Actions per Character (APC)

On average constructive and chorded methods yielded 2.58 ($SD = 0.13$) and 1.49 ($SD = 0.02$) APC, respectively. For the constructive method, APC started with 2.54 in session one and ended with 2.56 in session five. This effect was statistically significant ($F_{4,36} = 2.84, p < .05$). For the chorded method, APC started with 1.48 in session one and ended with 1.47 in session five. This result was also statistically significant ($F_{4,36} = 6.52, p < .0005$). Fig. 5b displays APC for both methods in all sessions.

### 14.3 Error Rate (ER)

Error rate (ER) is a commonly used error metric, which is traditionally calculated as the ratio of the total number of incorrect characters in the transcribed text to the length of the transcribed text [8]. On average, constructive and chorded methods yielded 6.05% ($SD = 10.11$) and 2.46% ($SD = 4.65$) ER, respectively. As expected, ER improved over the five sessions of testing for both methods. For the constructive method, ER dropped from 9.54% in session one to 3.37% in session five. This effect was statistically significant ($F_{4,36} = 2.86, p < .05$). For the chorded method, ER dropped from 3.42% in session one to 2.16% in session five. However, an ANOVA failed to identify a significant effect of session on ER for the chorded method ($F_{4,36} = 1.79, p = .15$). Fig. 6a

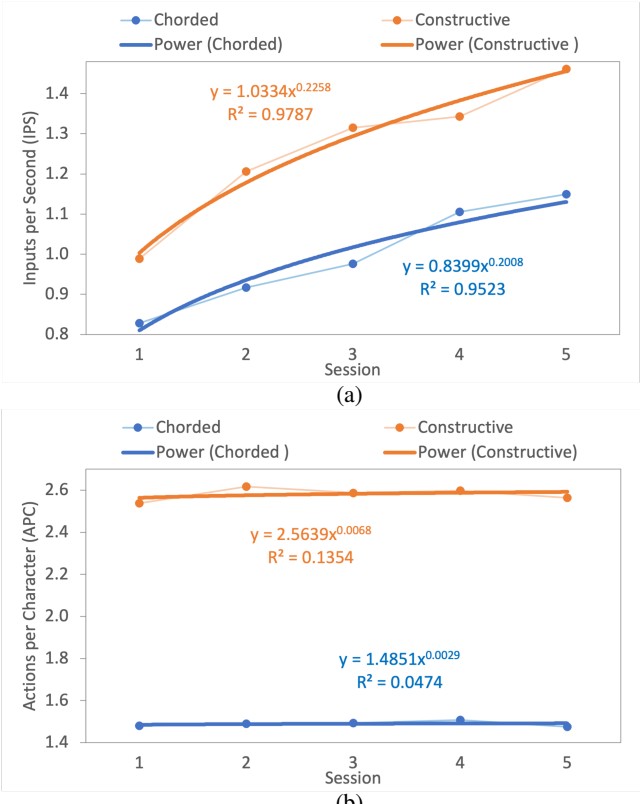

Figure 5: Average IPS for the two methods in all sessions (a) and average APC for the two methods in all sessions (b). Note the scale on the vertical axis.

displays ER for both methods in all sessions. Another widely used error metric, TER [39], yielded comparable result for the constructive method ($F_{4,36} = 3.33, p < .05$) and the chorded method ($F_{4,36} = 1.51, p = .22$).

### 14.4 Unit Error Rate (UnitER)

On average, UnitER for constructive and chorded methods were 1.32 ($SD = 2.2$) and 0.91 ($SD = 2.1$). For the constructive method, UnitER decreased from 2.04 in session one to 0.96 in session five. An ANOVA did not identify a significant effect of session on UnitER for constructive method ($F_{4,36} = 2.14, p = .09$). For the chorded method, UnitER started from 0.98 in session one and ended with 1.03 in session five. An ANOVA did not identify a significant effect of session on UnitER for chorded method as well ($F_{4,36} = 0.12, ns$). Fig. 6b displays UnitER for both methods in all sessions.

## 15 DISCUSSION

The IPS metric reflected a statistically significant increase in text entry speed across the five sessions for both input techniques. These results suggest that entry speed improved for both techniques with practice. The standard WPM [8] metric yielded similar statistical results. IPS per session also correlated well with the power law of practice [12] for both constructive ($R^2 = 0.98$) and chorded ($R^2 = 0.95$) methods (Fig. 5a).

There was a significant effect of session on APC for the two techniques. Similar trends were observed for UnitCX, and the commonly used KSPC metric. Though average values remain largely consistent (Fig. 5b), a Tukey-Kramer Multiple-Comparison test revealed that sessions 1 and 2 were significantly different for constructive, while

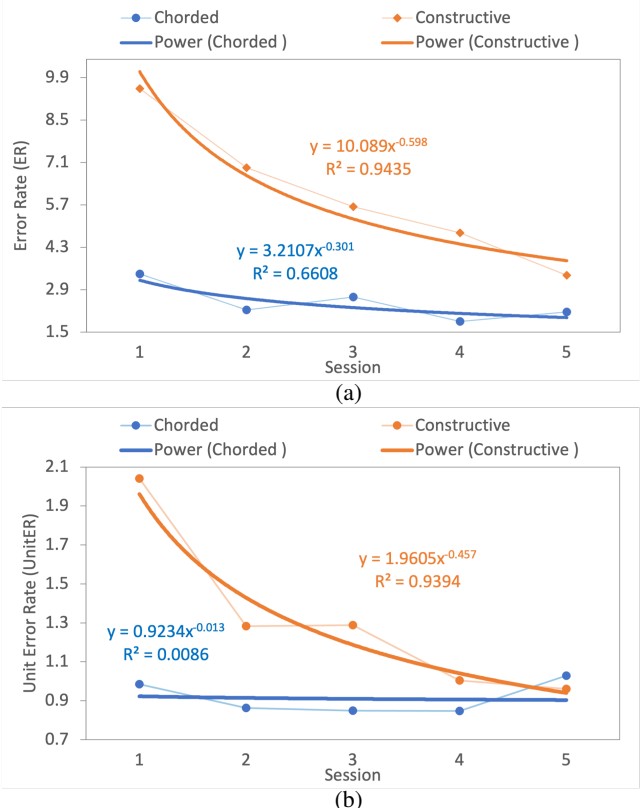

(a)

(b)

Figure 6: Average ER for the two methods in all sessions (a) and average UnitER for the two methods in all sessions (b). Note the scale on the vertical axis.

sessions 1, 4, and 5 were different for chorded. This finding may be due to a difference in group skewness or variance between sessions.

There was a significant reduction in ER from baseline to session five for the constructive technique (Fig. 6a). Accordingly, ER per session correlated well with the power law of practice [12] for the constructive method ($R^2 = 0.94$). Similarly, there was a reduction in UnitER from baseline to session five for the constructive method (Fig. 6b), though this difference was not statistically significant. UnitER per session also correlated well with the power law of practice [12] for the constructive method ($R^2 = 0.94$). The UnitER metric also provides useful additional details articulated in the following section. Neither ER or UnitER reflected a significant change across sessions for the chorded method. This finding, and the fact that IPS improved over each session for the chorded method, suggest that participants may have learned to type faster without a significant reduction in error rate; perhaps due to the physical nature of a chorded input technique. Additional data is needed to fully investigate this finding.

## 16 ACTION-LEVEL ANALYSIS

The above discussion demonstrates that the proposed metrics complement conventional metrics by facilitating a discussion of the methods' speed and accuracy in general. In this section, we demonstrate how the proposed metrics shed further light on these methods' performance and limitations.

To investigate which letters may have hampered text entry speed and accuracy, we first calculated UnitER for each letter. Because the letters do not appear the same number of times, we normalized the values based on each letter's total appearance in the study. We then identified the letters that were both difficult to enter and learn

to input accurately (henceforth "Not Learned"), and the letters that were difficult to enter but relatively easier to learn to input accurately (henceforth "Learned"). For this, we compared the average UnitER of each letter from the first three sessions to the last two sessions. The letters designated as "Learned" included the letters that demonstrated a significant improvement in UnitER from the first to the last sessions. In contrast, the "Not Learned" included the letters that consistently yielded higher UnitER in all sessions.

### 16.1 Constructive Method

Table 4 displays the top four Not Learned and the Learned letters from the Morse code keyboard. For better visualization, we calculated Unit Accuracy (UA) for these letters using simple transformation given by Eq. 8 and fitted them to power law of practice [12] to identify any trends (Fig. 7).

Fig. 7a illustrates the high inconsistency in UA across the Not Learned letters. However, for the Learned group (Fig. 7b), there is a prominent upward trend. Evidently, participants were learning these letters even in the last session, suggesting that performance with these letters would have continued to increase if the study lasted longer. We performed multiple comparisons between the letters to find the cause of entry difficulty, as well as the factors that facilitated learning. We identified the following trends that may have contributed towards the aforementioned trends.

The analysis revealed that participants were having difficulty differentiating between the letters that required similar actions to enter. For instance, it was difficult for them to differentiate between "h" ("...."") and "s" ("..."), and "k" ("-.-") and "c" ("-.-."), presumably, because their corresponding actions are very similar. A deeper analysis of the action-level errors revealed that participants frequently entered an extra dot when typing "k", resulting in the letter "c", and vice versa. This error totaled 23% of all UnitER for "c". Participants also made similar mistakes for other Not Learned letters. For example, they entered "s" ("...") instead of "h" ("....") and vice versa, resulting in 50% of all UnitER for "h" and 30% of all UnitER for "s". This information is vital. It helps the designer of a constructive method assign sufficiently different actions to frequent letters to avoid confusion, likely resulting in increased speed and accuracy.

Interestingly, participants tend to enter an extra dot or dash when these actions are repeated. For example, participants often entered "---.", "---.-" or similar patterns for z ("--.."), resulting in 17% of all UnitER in "z". Likewise, participants entered additional dots when entering g ("--."), such as z ("--.."), which resulted in 20% of all UnitER for "g". Similar errors were committed for other letters as well. These details are useful, since the designer can compensate by reducing the number of repeated actions for frequent letters.

### 16.2 Chorded Method

Table 5 displays the top four Not Learned and the Learned letters for the Senorita keyboard. Like the constructive method, we calculated UA for these letters using Eq. 8 and fitted them to power law of practice [12] to identify any trends, see Fig. 8. We observed that, like the constructive method, trends for all Learned letters were increasing, suggesting the occurrence of learning across the techniques. Participants were learning these letters even in the last session. This might

Table 4: Action-Level representation of the difficult to enter and learn (Not Learned) and difficult to enter but easier to learn letters (Learned) for the constructive method.

| Not Learned | Sequence | Learned | Sequence |
|---|---|---|---|
| z | --.. | g | --. |
| h | .... | p | .--. |
| s | ... | x | -..- |
| c | -.-. | q | --.- |

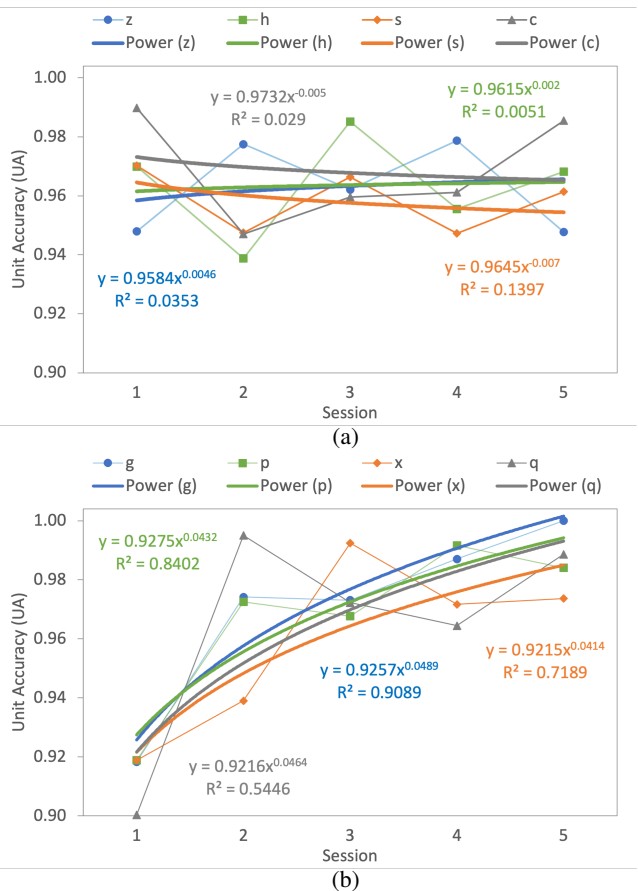

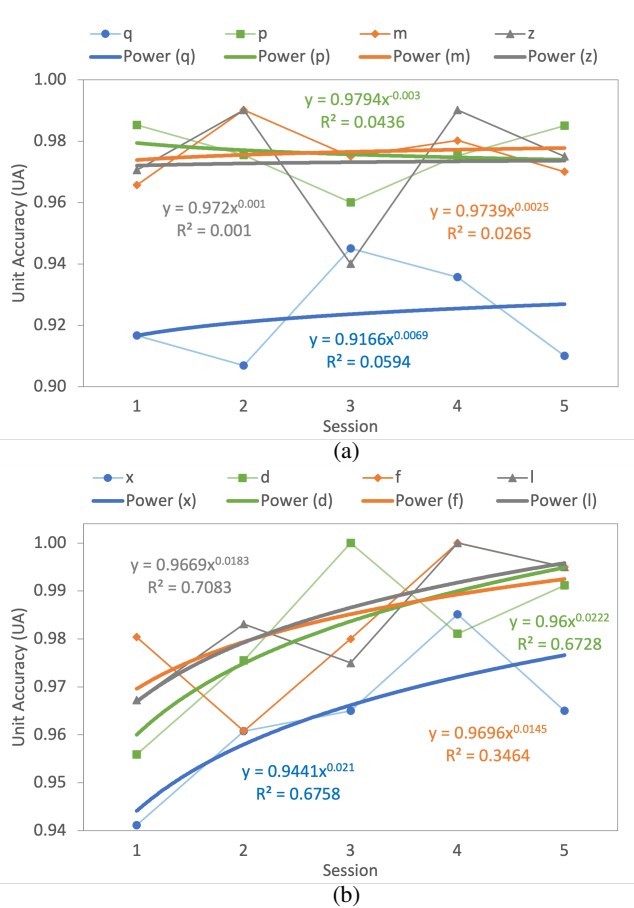

Figure 7: (a) Average Unit Accuracy (UA) of the letters "z", "h", "s", "c" per session with power trendlines. Trends for the letters show decreasing UA across sessions, indicating that learning was not occuring, (b) average UA of the letters "g", "p", "x", "q" per session with power trendlines. Trends for these letters show increasing UA as the sessions progressed, indicating learning.

device.

Figure 8: (a) Average Unit Accuracy (UA) of the letters "q", "p", "m", "z" per session with power trendlines. Trends for the letters show decreasing UA across sessions, indicating that learning was not occurring, (b) average UA of the letters "x", "d", "f", "l" per session with power trendlines. Trends for these letters show increasing UA as the sessions progressed, indicating learning.

indicate that the performance of these letters would have been much better if the study lasted longer. Multiple comparisons between action-level actions per letter identified the following trends that may have contributed towards the aforementioned trends.

Letters "q", "p", and "z" have the "r" key in their chords ("r", "er", and "rt", respectively), which is the furthest key from the right side. We speculate that it is difficult to learn, and be fast and accurate if letter has "r" in a chord pair, because "r" and "s" are the furthest letters from the edges (Fig. 9). Keys near the center of the screen are more difficult to reach than those at the edge. Relevantly, four letters with improving trendlines ("x", "d", "f", and "l") have chord pairs that are close to the screen edge. This detail may encourage designers to place the most frequent letters toward the edge of the

### 16.3 Time Spent per Letter

We compared the above analysis with the time spent to perform the sequences and chords for each letter in the last two sessions. The purpose was to further validate the metrics by studying if the letters that took additional time to enter correspond well to the Not Learned letters. Fig. 10a illustrates all difficult letters, highlighted in red (Not Learned) and green (Learned), identified for the constructive method. One can see the correlation between UnitER and the time spent to enter the sequence of these letters. This suggests that these letters require higher cognitive and motor effort to enter. Similarly, Fig. 10b illustrates all difficult letters, highlighted in red (Not Learned) and green (Learned), identified for the chorded method. One can see that the chords ("q, "v", "p", "g", "z", "j") required more time than taps ("s", "e", "n", "o", "r", "i", "t", "a"). However, the chords that are composed of the keys closer to the boundaries were much faster (e.g., "h", "u", "l", etc.). This further strengthens the argument of the previous section.

### 17 CONCLUSION

In this paper, we proposed a trio of action-level performance metrics (UnitER, UA, and UnitCX) aimed at multi-step text entry techniques

Table 5: Action-Level representation of the difficult to enter and learn (Not Learned) and difficult to enter but easier to learn letters (Learned) for the chorded method.

| Not Learned letter | Chord | Learned letter | Chord |
|---|---|---|---|
| q | *rs* | x | *ir* |
| p | *er* | d | *eo* |
| m | *en* | f | *io* |
| z | *rt* | l | *at* |

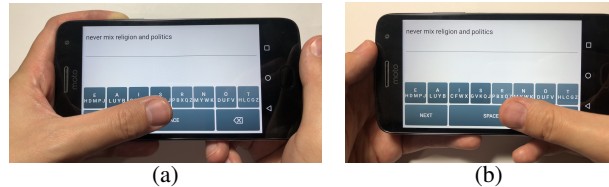

(a)                    (b)

Figure 9: The user stretching her thumb to reach the letter "s" (a) and the letter "r" (b).

that account and compensate for the constructive and chorded input process. We validated these metrics in a longitudinal study involving one constructive and one chorded technique. In our presentation, we used existing techniques such as Morse code [19, 38] and Senorita [31], not to change their mapping or design, but to amply demonstrate how can the newly proposed metrics be applied for different multi-step techniques and to give a deeper insight into possible limitations of the techniques with an emphasis on learning. None of the metrics would give the conclusions automatically, but they could point towards limitations. Our UnitER helped us to investigate specific erroneous characters, while conventional metrics failed to identify them. The results of this study demonstrate how the proposed metrics provide a deeper understanding of action-level error behaviors, particularly the difficulties in performing and learning the sequences and chords of the letters, facilitating the design of faster and more accurate multi-step keyboards. Although there was previously no formal method to analyze action-level actions of the multi-step method, it is likely that veteran text entry researchers perform similar analysis on user study data. This work provides a formal method that will enable researchers new to the area to perform these analyses, and facilitate better comparison between methods from the literature.

## 18 FUTURE WORK

We intend to combine the insights gained from the proposed action-level metrics, particularly the most error-prone characters and the types of errors frequently committed in the sequence or chord of these characters, with machine learning approaches to make multi-step keyboards adapt to human errors. Such a system can also use language models to provide users with more effective auto-correction and word suggestion techniques.

### ACKNOWLEDGMENTS

We thank Anna-Maria Gueorguieva for helping us with the user study. This project was supported in part by the Graduate Student Opportunity Program (GSOP) Fellowship at UC Merced.

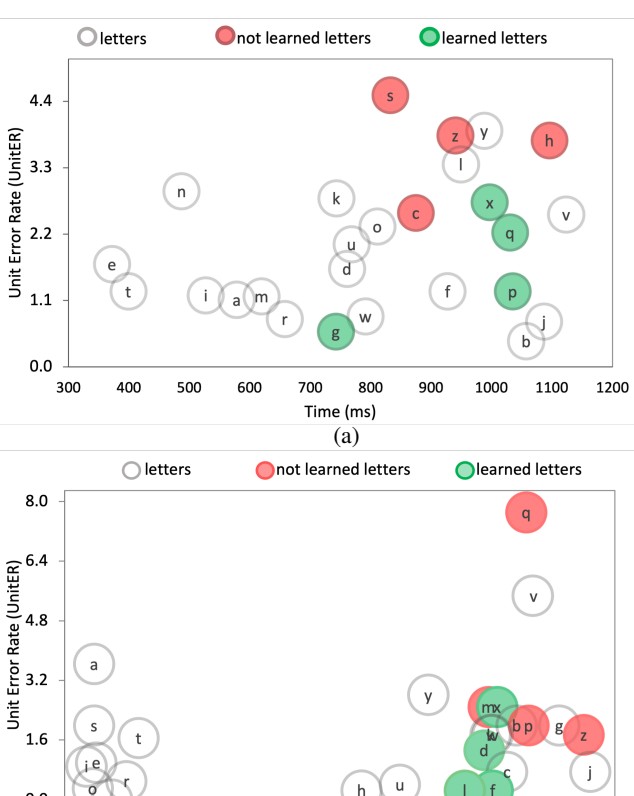

Figure 10: (a) Average UnitER of all letters vs. the time spent to perform the sequence for those letters for constructive method, (b) average UnitER of all letters vs. the time spent to perform the chords for those letters with the chorded method.

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

3300866
