# OpenReview forum: "Using Action-Level Metrics to Report the Performance of Multi-Step Keyboards"
_graphicsinterface.org/Graphics_Interface/2021/Conference — GI 2021_

### Official Review · AnonReviewer1 · 2021-01-06
**New performance metrics for text entry techniques that are either constructive or chorded.**

**Rating:** 6
**Confidence:** 2

**Review:**

This paper proposes new performance metrics for text entry techniques that are either constructive or chorded.  The paper validates the metrics by showing how they helped the authors analyze data obtained from a longitudinal study comparing a specific constructive technique and a specific chorded technique.

This is a focused submission, contributing metrics that have the potential to interest other researchers working on chorded or constructive text-entry techniques.  The paper makes a good case for the limitations of existing metrics in that they cannot capture how far away user actions are from the correct entry.  This in turn limits a researcher’s ability to see whether user performance is improving with practice, even if they are not yet getting the actions completely “right”.

In terms of validating the metrics, the paper’s approach seems reasonable, but perhaps not overly strong.  The authors show how they used their metrics with their own data to draw conclusions about the techniques studied. It would be useful to reflect on the limitations of this validation approach and how others have validated text entry metrics in prior work.  For example, have arguments been primarily analytical, have other papers solicited feedback from researchers outside of the team, etc.?

The contribution seems thin for the length of the paper.  This is partly owing to a fair amount redundancy in the first four pages.  Regardless of the final decision for the paper, I would recommend restructuring and streamlining the first three sections to ensure that the motivation is presented only once, followed by a related work with a clear discussion of the gap in the literature that this work is filling.  This could then transition directly into the metrics and notation.

In terms of paper length, it also is not clear that the study needs to be described to the extent that it is if its main purpose is to demonstrate how the metrics were used on the data.  It appears there was a confound in the study in that participants had more prior experience with the constructive method than with the chorded method.  As such, it is not clear that seeing the statistical comparisons between the two techniques is adding value to the paper.

A few other presentation issues that impact the paper’s clarity:

- For a more general HCI audience, I would recommend defining “constructive” and “chorded” text entry methods early in the paper.
- Table 1 does not help showcase why UnitER is an improvement ove ER.  It is true that the numbers are different, but without a more informative caption, it is not clear why this difference is meaningful.
- There is notation in 6.1.1 that I did not see introduced in the “Notation” section. (e.g. a[1: ])

Given the venue, I am mildly supportive of the paper, but it is not a huge leap forward and the presentation could be improved.  To contextualize my assessment, I am an HCI researcher, but not an expert in text entry techniques nor in their validation.

---

### Official Review · AnonReviewer2 · 2021-01-12
**New action-level performance metrics that account and compensate for the different input methods of constructive and chorded input processes.**

**Rating:** 6
**Confidence:** 2

**Review:**

This paper introduced new action-level performance metrics that account and compensate for the different input methods of constructive and chorded input processes. The results from a user study demonstrate how the proposed metrics can be applied to different multi-step techniques.

In general, the paper uses a sound approach. However, I believe there is some room for improvement. Below are lists of things I would like to see the authors address or clarify:

- Section 2, the authors discussed some accuracy and performance metrics but did not mention throughput metrics [https://doi.org/10.1145/3290605.3300866]

- Section 2, the authors state that accessibility text entry systems mainly use constructive or chorded techniques, but there is little discussion about each technique. I would suggest adding a description about each technique and how they are similar/different, at least in broad terms.

- Section 6.3: The authors mentioned that there are “… more sophisticated methods are available in the literature” than UnitCX. I would suggest adding a paragraph in related work about the more sophisticated methods to measure inputs unit’s complexity.

- Section 7: Why did the authors decide to do a comparative study between a constructive and a chorded text entry technique? How does this approach support their motivation of introducing detailed metrics of multi-step keyboard interaction that can facilitate improved input method design? This should be explained better.

- Section 11: The authors should discuss how their participant sampling affected the study, as none of the participants had prior experience with any chorded methods, but eight had used a constructive method in the past.

- Section 13: Why did the authors decide to repeat the same phrases instead of introducing new phrases each time? Users might thus memorize the sequence of the whole phrase. How could this affect the outcome of the study?

- Section 14.2: Action per Character (APC) should have been introduced before being used, e.g., in section 4 or 5.

- Section 15: The discussion lacks citations of other work to relate the outcomes to the field.

- Section 16: Why did the authors decide to first present the discussion in section 15 and then the Action-Level Analysis as section 16? Also, how does the analysis of the metrics help to inform designers to develop new action sequences beside “encourage designers to place the most frequent letters toward the edge of the device” which is well-known and has been studied in other research, e.g., [https://doi.org/10.1145/2702123.2702439]?

- Clarity of Presentation: Overall, the presentation of the paper is clear. I would suggest improving the presentation of section 16 since the figures can be hard to follow.

- Technical soundness and completeness: I believe this system can be implemented by a competent graduate student. After all, the core technology part uses existing applications, and only metrics need to be calculated.

- References: Appears adequate.

Most of the above are minor comments that can be addressed in the camera-ready version. In my opinion, although some findings might seem unsurprising, the paper has enough contribution. I would suggest accepting this paper.

---

### Official Review · AnonReviewer3 · 2021-01-12
**Small but potentially useful contribution**

**Rating:** 6
**Confidence:** 3

**Review:**

This paper proposes three new metrics for assessing performance of multi-step text entry methods. The work has interesting potential for application in this domain, and multi-step entry methods are important, particularly for accessibility. The work is reasonable, but has a few weaknesses that limit its contribution

**Framing.** The metrics are presented as performance metrics and positioned as an alternative to more traditional measures of speed and accuracy. But the measures actually reflect learning. They do not capture the end product but rather the process that the user took to get to that final product. This isn’t to say that looking at learning isn’t important, just that the paper lacked grounding in a discussion of it and how measuring it is typically tackled in multi-step evaluation work. For example, later in the paper one of the demonstrated uses of the new metrics was to aid in the identification of common character substitutions. Finding pairs of characters that are commonly swapped is fairly common. I imagine this work facilitates identification of them or presents a more systematic way of examining them, but it’s hard to know exactly how without such a discussion of prior work.

**Weak statistical results.** It was disappointing that significant results were not found for the new metrics over the course of the study. Particularly as measures of learning, this seems like a fairly important bar to pass. The paper does demonstrate some utility without that significance, but it leaves the reader questioning how strong these metrics are. Relatedly, the fact that there were other significant differences that the paper couldn’t explain leaves me wondering if there isn’t a problem in the data and if perhaps more work is needed to examine it in depth.

**Ties to accessibility.** This is a small and easily addressable point but I think it’s important. The research uses two accessible keyboards for the context of the study, but does not use assistive technology users in the experiment. I think this is reasonable given the purpose of the research which is to test the metrics and not the keyboards themselves. The paper even cautions the reader to not consider the results as indications of how to adapt or modify the keyboards (“…, not to change their mapping or design…”, Conclusion). This is great, but I think it should be stated more clearly that the specific results (i.e., which characters were problematic, confused, took longer to learn, etc.) would likely be different if the study were repeated with representative users, and thus the findings here should not be interpreted as recommendations for updating or changing the keyboard designs. Otherwise, I see some potential for harmful misinterpretation of the these results by future readers.

Overall, this is a relatively modest paper, presenting a small contribution. The framing of the work and depth of the data analysis could be improved to strengthen the results, but it makes an acceptable contribution in its current form. I could see the work being useful to people working in this space. I remain ambivalent about its acceptance, and accordingly provide a weak vote in favour of its acceptance.

---

### Meta-Review · Area_Chair1 · 2021-01-16

**Recommendation:** Accept
**Confidence:** 3

**Metareview:**

This paper introduces metrics for assessing the performance of constructive and chorded input processes. The reviewers agree that the work has potential for applications in the text entry and accessibility domain.

R1 would like a clear reflection on the limitations of the validation approach and how prior work has validated similar metrics. This is echoed by R2 and R3 who would like to see the authors address how measuring performance and learning is typically tackled in multi-step evaluation work.


  Summary and Required Changes:
- Restructure the first three sections to ensure that the motivation is presented once, followed by a related work with a clear discussion of the gap in the literature.
- Add the throughput metric to related work [https://doi.org/10.1145/3290605.3300866 ]
- Add a description about constructive or chorded technique
- Add a paragraph in related work about the more sophisticated methods to measure inputs unit’s complexity
- Discuss how does a comparative study support their motivation
- Discuss how their participant sampling affected the study
- Relate the study outcomes to the other work in the field

Despite these required changes, all reviewers vote in favour of its acceptance. I suggest that the authors take care to add the missing information regarding the validation approach and related work. Overall, with these revisions, I recommend accepting this paper.

---

### Decision · Program_Chairs · 2021-01-16

Accept